# Protein and Macronutrient Metabolism in Liver Cirrhosis: About Sarcopenia

**DOI:** 10.3390/nu17213346

**Published:** 2025-10-24

**Authors:** Seul Ki Han, Soon Koo Baik, Moon Young Kim

**Affiliations:** 1Department of Internal Medicine, Yonsei University Wonju College of Medicine, Wonju 26426, Republic of Korea; 2Regenerative Medicine Research Center, Yonsei University Wonju College of Medicine, Wonju 26426, Republic of Korea; 3Cell Therapy and Tissue Engineering Center, Yonsei University Wonju College of Medicine, Wonju 26426, Republic of Korea

**Keywords:** chronic liver disease, malnutrition, sarcopenia, branched-chain amino acids, macronutrient metabolism, hepatic encephalopathy, ammonia

## Abstract

Malnutrition, sarcopenia, and frailty are highly prevalent in patients with chronic liver disease and are closely associated with poor clinical outcomes. This review highlights the complex interplay between macronutrient metabolism and muscle wasting in liver cirrhosis. We explore the alterations in glucose, lipid, and amino acid metabolism that occur in cirrhosis, including the role of skeletal muscle in compensatory ammonia detoxification. The review also discusses the emerging evidence on sarcopenia as a prognostic marker and therapeutic target, with a focus on the role of branched-chain amino acid (BCAA) supplementation. While several studies have demonstrated the clinical benefits of BCAA in improving muscle mass, hepatic encephalopathy, and quality of life, results remain mixed, emphasizing the need for further well-designed clinical trials. Understanding the muscle–liver–gut axis may offer novel insights into therapeutic strategies for managing sarcopenia in liver disease.

## 1. Introduction: Malnutrition in Chronic Liver Disease

Malnutrition, sarcopenia, and frailty are highly prevalent in patients with cirrhosis and hepatocellular carcinoma, affecting 40–70%, 30–40%, and 18–43% of patients, respectively [1,2,3,4]. In particular, sarcopenia, defined as the loss of skeletal muscle mass, strength, and function, is increasingly recognized as a major prognostic determinant in cirrhosis [5]. These complications are strongly associated with increased risk of infection, hepatic encephalopathy (HE), hospitalization, poor treatment response, and mortality [4,5,6,7]. Multiple mechanisms contribute to malnutrition in liver disease, including reduced dietary intake due to anorexia, early satiety, or ascites; altered carbohydrate metabolism with insulin resistance, chronic inflammation, hyperammonemia; Treatment-related factors, including the use of diuretics, alcohol abstinence, surgery, and cancer therapy can also contribute to malnutrition in patients with liver disease by further aggravating metabolic and nutritional imbalances [6,8,9].

Several factors contribute to malnutrition in patients with chronic liver disease. The first and most common is inadequate intake. This may result from early satiety, anorexia, nausea and vomiting, dysgeusia, unpalatable dietary restrictions (e.g., low sodium or low potassium diets), impaired level of consciousness, fluid restriction, and frequent fasting related to procedures and hospitalizations, all of which predispose patients to insufficient oral intake under a wide range of circumstances [6,9,10].

Micronutrient deficiencies can also exist in patients with cirrhosis, and this varies depending on the etiology. In patients with alcohol-related liver disease, deficiencies in folate, thiamine, zinc, selenium, and vitamins D and E are characteristic [11]. In contrast, patients with cholestatic liver disease are frequently affected by deficiencies in fat-soluble vitamins due to impaired bile function. Beyond intake-related factors, impaired nutrient absorption is another major contributor, driven by malabsorption, maldigestion, and altered macronutrient metabolism [12].

Micronutrient deficiencies can further aggravate cirrhosis-related symptoms and complications. For example, vitamin D deficiency contributes to increased energy expenditure and impaired muscle function [13], zinc deficiency worsens ammonia detoxification and is associated with hepatic encephalopathy (HE) and sarcopenia [14], and magnesium deficiency correlates with cognitive decline and muscle weakness [15].

In addition to these mechanisms, alterations in enterohepatic circulation of bile salts, portosystemic shunting, pancreatic enzyme deficiency, bacterial overgrowth, dysbiosis of intestinal flora, and enteropathy are all factors linked to malnutrition in patients with chronic liver disease [6,16]. This review primarily addresses the macronutrient aspects of malnutrition in chronic liver disease, with a particular emphasis on amino acid dysregulation-induced sarcopenia and the therapeutic approaches aimed at restoring metabolic balance.

## 2. The Importance of Nutritional Assessment in Chronic Liver Disease

Nutrition plays a central role in the management of chronic liver diseases. Current recommendations suggest an energy intake of 25–35 kcal/kg/day [17,18] and protein intake of 1.2–1.5 g/kg/day, while protein restriction is not advised as it may worsen HE and protein catabolism accelerated by a low-protein diet [17]. Late evening snacks (LESs) improve nitrogen balance and lean mass, although the survival benefit remains inconsistent [19,20].

Early nutritional intervention is particularly crucial in decompensated cirrhosis, where metabolic derangements are more pronounced. Adequate nutritional support, defined as a total caloric intake of at least 22 kcal/kg/day, has been demonstrated to confer prognostic benefit in hospitalized patients with alcoholic hepatitis [7,18]. Adequate caloric (25–35 kcal/kg/day) and protein support (1.2–1.5 g/kg/day) help mitigate nitrogen imbalance [17]. Moreover, avoidance of unnecessary protein restriction in hepatic encephalopathy is now emphasized [6,9,21]. And one study demonstrated that implementing appropriate nutritional intervention through counseling for hospitalized patients with cirrhosis reduced the three-month readmission rate by more than 25% [22]. Nutritional imbalance—manifesting as malnutrition and sarcopenia—has been consistently associated with adverse clinical outcomes and increased mortality in patients with liver disease [4,5,23]. Based on these findings, current clinical guidelines do not recommend protein restriction in patients with liver disease. Instead, they advocate for a daily protein intake of 1.2–1.5 g/kg body weight and a total energy intake exceeding 30 kcal/kg body weight per day [9,21]. In patients with advanced cirrhosis complicated by ascites or hepatic encephalopathy, recommendations increase to up to 40 kcal/kg/day to meet heightened metabolic demands [6,9,21] (Table 1). Therefore, comprehensive nutritional assessment and management constitute a critical component of care in this population.

## 3. Macronutrients Metabolism and Beyond Malnutrition

Cirrhosis is characterized by profound alterations in energy metabolism, leading to accelerated protein catabolism and muscle wasting. Patients with advanced liver cirrhosis exhibit a state resembling chronic starvation, with reduced glycogen reserves, impaired gluconeogenesis, and increased lipid and amino acid oxidation [14,24]. These metabolic shifts are compounded by hyperammonemia, systemic inflammation, and hormonal dysregulation, all of which contribute to skeletal muscle loss and the development of sarcopenia.

(1)Glucose metabolism

As cirrhosis progresses, resting energy expenditure (REE) increases, reflecting a metabolic adaptation characterized by a switch in the primary energy substrate from glucose to fatty acids [12,14]. Glucose metabolism in the liver is orchestrated by the interplay of glycogenesis, glycogenolysis, glycolysis, and gluconeogenesis. Glycogen metabolism is tightly regulated through reciprocal actions of glycogen synthase and glycogen phosphorylase [25].

Hepatic glycolysis is controlled by rate-limiting enzymes such as glucokinase, phosphofructokinase-1 (PFK-1), and liver-type pyruvate kinase (L-PK) [26]. These enzymes are transcriptionally controlled by sterol regulatory element binding protein-1c (SREBP-1c) and carbohydrate response element binding protein (ChREBP), which integrate insulin and glucose signals to couple glycolysis with lipogenesis [26]. During fasting, gluconeogenesis becomes the predominant pathway to sustain systemic glucose supply. Gluconeogenesis is an energy-expensive procedure that may further increase resting energy expenditure (REE) in these patients. Recent studies have demonstrated that hepatic glucose production in patients with steatohepatitis is significantly increased in those with greater degrees of hepatic inflammation, ballooning, and fibrosis, but not in relation to steatosis itself [27]. These findings indicate that intrahepatic inflammation and fibrotic remodeling, rather than lipid accumulation per se, represent the key drivers of metabolic dysregulation and glucose homeostasis impairment in MASH.

As a result of these mechanisms, patients with cirrhosis exhibit reduced glycogen stores, impaired gluconeogenesis, and an increased level of insulin. One study demonstrated that even cirrhotic patients with normal Glycated hemoglobin (HbA1c) levels have elevated insulin resistance and fasting glucagon levels, along with reduced incretin effect, resulting in impaired glucose disposal compared to healthy individuals [28]. Furthermore, a study comparing cirrhotic patients to healthy controls using oral glucose tolerance tests showed that insulin resistance correlates positively with the severity of cirrhosis, as reflected by higher Child–Pugh (CTP) grades [29].

During the fasting periods of 2 to 6 h, hepatocytes maintain systemic glucose availability primarily through glycogenolysis, in which stored hepatic glycogen is degraded to glucose. With prolonged fasting, however, hepatic glycogen stores become depleted, and hepatocytes shift toward gluconeogenesis, synthesizing glucose de novo from alternative substrates such as lactate, amino acids, and glycerol to sustain energy homeostasis [30]. In this process, a high incidence of hypoglycemia is frequently observed in patients with liver cirrhosis. Pfortmueller reported hypoglycemia per se predictive marker for lower estimated survival compared with high-glycemic or normoglycemic cirrhosis patients [31].

In advanced liver disease, disturbances in glucose homeostasis—characterized by insulin resistance and episodes of hypoglycemia—represent major metabolic derangements that can directly influence patient survival. These abnormalities reflect both impaired hepatic glucose regulation and systemic metabolic dysfunction, underscoring their critical role in the prognosis of patients with decompensated cirrhosis. These mechanisms highlight that dysregulation of hepatic glucose metabolism is not only a consequence of impaired enzyme activity but also reflects altered transcriptional programs and hormone signaling.

(2)Lipid metabolism

Intrahepatic lipid accumulation is commonly observed in patients with chronic liver disease, including those with metabolic dysfunction such as MASLD and diabetes [32,33,34], as well as in individuals with viral hepatitis [35,36] or alcoholic hepatitis [37]. Increased intrahepatic fat content is rarely encountered in cholangiopathies such as primary biliary cholangitis (PBC) and primary sclerosing cholangitis (PSC) [38]. Hepatic steatosis usually primarily from enhanced de novo lipogenesis (DNL) and excessive fatty acid transport, driven by increased adipose tissue lipolysis, insulin resistance, and high dietary intake of saturated fatty acids [39]. Within the liver, these substrates are converted into triglycerides and other complex lipid species. This process is regulated by key transcription factors such as SREBP-1c and carbohydrate-responsive element binding protein (ChREBP), as well as enzymes like acetyl-CoA carboxylase (ACC), which collectively serve as central regulators of fatty acid synthesis [40]. In addition, mitochondrial dysfunction and impaired metabolic flexibility have been implicated as key contributors to lipid dysregulation in MASH [26]. Patients with MASH have shown increased hepatic fatty acid oxidation and elevated levels of beta-hydroxybutyrate, particularly among women [41].

However, in patients with advanced cirrhosis resulting from progressive steatotic liver disease, the fasting state is characterized by elevated levels of non-esterified fatty acids (NEFAs), high rates of fatty acid turnover and oxidation, and hyperinsulinemia [42]. This reflects the previously described resistance of adipose tissue lipolysis to insulin and is further supported by the elevated levels of free fatty acids, indicating a metabolic shift toward alternative energy sources beyond glucose [43]. Increased energy demands in decompensated liver cirrhosis are associated with elevated lipid mobilization, a mechanism that is closely interconnected with the immune response [44]. Lipids interact to trigger inflammation and several lipid derivatives are important inflammatory mediators that act as intracellular signaling molecules [45,46]. Variations in the serum ceramides are associated with hepatic decompensation and survival rate in cirrhosis [47], and low levels of sphingosine-1-phosphate are predictive of increased mortality [48]. Similarly, altered sphingolipid metabolism characterizes malnutrition in hospitalized individuals with decompensated cirrhosis [49]. In addition, free fatty acid–related alterations have been observed, including a progressive decrease in total polyunsaturated fatty acids (PUFAs) as liver disease advances [50,51]. There is also an increase in the ratio of arachidonic acid (AA) to eicosapentaenoic acid (EPA), along with changes in lipid mediators such as leukotriene E and prostaglandin E [50,52]. However, further studies are needed to clarify the clinical significance and mechanistic implications of these findings.

(3)Amino acid metabolism

Amino acid metabolism can be changed in chronic liver disease. For example, glucogenic amino acids serve as substrates for gluconeogenesis [53], while dietary proteins and amino acids may also contribute to de novo lipogenesis, facilitated by an enhanced tricarboxylic acid cycle activity [54]. Many amino acids are considered substrates for gluconeogenesis, precursors for neurotransmitters, and regulators of cellular signaling pathways [26]. In the early stage of liver cirrhosis, known as compensated cirrhosis, systemic ammonia levels are typically maintained within the normal range despite progressive impairment of hepatic urea cycle function [55,56]. This is primarily due to a compensatory increase in ammonia detoxification via glutamine synthesis in skeletal muscle [14]. In healthy individuals, ammonia is metabolized equally by the hepatic urea cycle and the muscle-based glutamine-synthesizing system. However, as hepatic function declines, skeletal muscles assume a greater role in maintaining ammonia homeostasis. This shift leads to increased utilization of branched-chain amino acids (BCAAs), resulting in reduced plasma BCAA levels and a decline in the Fischer ratio or BCAA-to-tyrosine ratio (BTR), a metabolic hallmark of compensated cirrhosis [57]. These metabolic changes lead to a 15–30% increase in energy requirements [24], which can be aggravated with the insufficient dietary intake resulting from cirrhosis-related complications such as ascites, HE, and dysgeusia, a significant malnutrition-contributing factor [58].

In normal liver physiology, branched-chain amino acids (BCAAs)—including leucine, isoleucine, and valine—are not primarily metabolized in the liver but rather in skeletal muscle, where they serve as substrates for energy production and regulators of protein synthesis. Within muscle tissue, branched-chain amino acid transaminase (BCAT) catalyzes the conversion of BCAAs to their corresponding α-keto acids, which subsequently enter the tricarboxylic acid (TCA) cycle through the action of the branched-chain α-keto acid dehydrogenase complex (BCKDH) [59].

In cirrhosis, several pathological alterations in amino acid metabolism become evident: (1) Aromatic amino acids (AAAs; phenylalanine, tyrosine, and tryptophan) increase, whereas BCAAs decrease. This imbalance results from impaired hepatic detoxification and metabolic dysfunction, leading to AAA accumulation, while BCAAs are excessively consumed in skeletal muscle as alternative energy substrates [60,61]. (2) Reduced hepatic ammonia clearance drives skeletal muscle to compensate by fixing ammonia into glutamine, a process that consumes α-ketoglutarate derived from BCAA catabolism, further depleting BCAA levels [61]. (3) Leucine normally activates the mTORC1 signaling pathway to promote muscle protein synthesis; however, its reduction in cirrhosis suppresses mTORC1 activity and exacerbates sarcopenia [61,62,63] (4) Isoleucine, which contributes to fatty acid β-oxidation and gluconeogenesis, becomes less available as these metabolic pathways are downregulated in hepatic dysfunction, resulting in impaired glucose production and reduced lipid oxidation [63].

Collectively, these disturbances produce a signaling imbalance characterized by decreased BCAA and increased AAA levels, which inhibit muscle insulin signaling and peripheral glucose utilization, perpetuating systemic insulin resistance. Such alterations are reflected in the plasma amino acid profile, represented by the Fischer ratio ([BCAA]/[AAA]) [60]. Under normal conditions, this ratio ranges from 3.0 to 3.5, whereas in cirrhosis, it often falls below 1.0–2.0 due to elevated AAA concentrations [64].

Thus, alterations in amino acid metabolism can serve as a useful biochemical marker for assessing both nutritional status and hepatic functional impairment in patients with advanced liver disease. This biochemical pattern is closely associated with an increased risk of hepatic encephalopathy [64,65].

Meanwhile, alterations in intestinal microbiota also contribute to ammonia dynamics [66]. As liver disease progresses, compositional changes in the gut flora occur, including the emergence of urease-producing species such as *Streptococcus salivarius*, which are not typically present in healthy individuals [67]. These bacteria enhance intestinal ammonia production, thereby increasing the burden on extrahepatic detoxification systems. Although the liver’s detoxifying capacity is compromised, the compensatory response in skeletal muscle prevents hyperammonemia during this compensated phase. This adaptive interplay between altered gut microbiota and skeletal muscle metabolism underpins the subclinical metabolic shifts observed before overt hepatic decompensation [58].

Beyond amino acid imbalance, cirrhosis is characterized by a profound disturbance in protein metabolism. Decreased hepatic synthesis of albumin and fibrinogen, along with increased muscle proteolysis via the ubiquitin–proteasome system, contributes to progressive protein-energy malnutrition [68]. Chronic inflammation and hyperammonemia suppress mTORC1 signaling and hepatic IGF-1 production, further impairing anabolic responses and accelerating sarcopenia [62].

## 4. Malnutrition Associated with Cirrhosis: Sarcopenia

As liver disease progresses from the compensated to the decompensated stage, hepatic protein synthesis capacity, particularly albumin production, declines, and hypoalbuminemia serves as a reliable predictor of poor prognosis in liver cirrhosis [5,12,69]. With disease progression, impaired hepatic ammonia detoxification shifts the burden to skeletal muscle, where ammonia is metabolized via glutamine synthesis [70]. As a consequence, autophagic proteolysis and the expression of myostatin, which is a key inhibitor of muscle protein synthesis, are upregulated in cirrhosis [71]. Simultaneously, the ongoing demand for BCAAs to support residual glutamine synthesis further depletes these essential amino acids, compounding protein-energy malnutrition and impairing albumin synthesis [71]. These alterations are closely linked to progressive sarcopenia, driven by chronic inflammation, nutritional deficiencies, and hormonal dysregulation [26]. Clinically, this transition is marked by the onset of hyperammonemia, hypoalbuminemia, and neuropsychiatric manifestations such as hepatic encephalopathy [12,14]. These features define the decompensated phase of cirrhosis, where both hepatic and extrahepatic mechanisms of ammonia detoxification are overwhelmed [14]. Sarcopenia and impaired ammonia metabolism reinforce one another, forming a vicious cycle in which muscle loss reduces ammonia clearance capacity, thereby exacerbating hyperammonemia and related complications. Meta-analyses demonstrate that sarcopenia approximately doubles the risk of mortality in cirrhotic patients [72]. Mechanistically, sarcopenia in cirrhosis is driven by a combination of insulin resistance, systemic inflammation, hyperammonemia-induced muscle breakdown, and mitochondrial dysfunction [6]. As sarcopenia is now recognized as an independent predictor of morbidity and mortality in cirrhosis, its pathophysiological underpinnings highlight the need for early detection and targeted interventions.

## 5. Sarcopenia Management: Evaluation and Treatment

In 2021, the American Association for the Study of Liver Diseases (AASLD) issued guidelines for sarcopenia [6], which adopted the definitions of malnutrition, frailty, and sarcopenia previously used in geriatrics, but operationally reinterpreted these terms in the context of cirrhosis and described them at the beginning of the guideline. Various modalities, including anthropometric measurements, bioelectrical impedance analysis (BIA), dual-energy X-ray absorptiometry (DEXA), ultrasound, and magnetic resonance imaging (MRI) have been investigated for the assessment of sarcopenia [21]. However, computed tomography (CT) imaging is currently the gold standard for assessment of muscle mass in cirrhosis. Despite its accuracy, the high cost and exposure to ionizing radiation limit the routine use of CT for the purpose of detecting sarcopenia impractical in many clinical settings. Muscle mass is conventionally reported as the skeletal muscle index (SMI), calculated as the total skeletal muscle area at L3 normalized to height [73,74]. And the level of SMI  <  50 cm^2^/m^2^ for men and  <39 cm^2^/m^2^ for women is used to define sarcopenia in patients with end-stage liver disease awaiting LT [73].

Therapeutic approaches for sarcopenia include adequate protein intake (1.2–1.5 g/kg/day), late evening snacks [75], and BCAA supplementation [76]. Exercise interventions improve physical performance and quality of life. In a prospective study involving patients with cirrhosis categorized as Child–Pugh class B or C and a mean MELD-Na score of 17, a structured exercise regimen was implemented consisting of moderate-intensity activity for 30 min per day, five days per week, averaging 100 steps per minute and 5000–7000 total daily steps. This intervention led to potential improvements in appendicular lean mass index (*p* = 0.07) and lower extremities lean mass index (*p* = 0.06) [77]. Furthermore, a meta-analysis evaluating the effects of exercise in patients with cirrhosis found that exercise training significantly improved 6-min walking distance, a key indicator of sarcopenia. Notably, in a subgroup analysis of patients who underwent combined aerobic and resistance exercise, the incidence of serious adverse events (SAEs) was 6.25% (7/112) compared to 24.7% (18/73) in the control group, indicating an approximate 18% absolute reduction in SAE rates [78].

L-ornithine L-aspartate (LOLA) is a compound consisting of the amino acid ornithine and aspartate. Ornithine is known to activate the urea cycle in the liver, while aspartate facilitates glutamine synthesis. In a rat model, administration of LOLA was shown to reduce plasma and muscle ammonia levels, increase lean body mass, and lead to visible improvements in skeletal muscle mass and fiber diameter. Additionally, LOLA treatment was associated with reduced expression of myostatin, suggesting that ammonia-lowering therapy may represent a potential strategy for addressing sarcopenia [62].

In patients with cirrhosis and reduced testosterone levels, administration of testosterone, an anabolic steroid, was evaluated in individuals with FibroScan ≥ 15 kPa (F4 stage cirrhosis) over a 52-week period. The intervention resulted in increased muscle mass, decreased fat mass, and improvements in glucose metabolism. Furthermore, significant gains in total bone mass and bone mineral density at the femoral neck were observed, along with a modest improvement in overall survival [79].

More recently, terlipressin, a vasoconstrictor acting on vasopressin receptors, was investigated in a cohort of 30 patients with decompensated cirrhosis, ascites, and sarcopenia. After 12 weeks of treatment, patients showed improved handgrip strength (+3.09 kg; 95% CI: 1.11–5.08; *p* = 0.006), a reduction in cumulative ascitic drainage volume (−11.39 L; 95% CI: 2.99–19.85; *p* = 0.01), and a decrease in the number of large-volume paracenteses (−1.75; 95% CI: 0.93–2.59; *p* < 0.001). These findings suggest that terlipressin may contribute to improved quality of life in patients with decompensated cirrhosis [80].

## 6. Evidence from RCTs and Systematic Reviews on BCAA Supplementation

Branched-chain amino acids (BCAAs) have been extensively investigated as a therapeutic adjunct for sarcopenia in cirrhosis. In cirrhotic patients, diminished hepatic gluconeogenesis shifts ammonia detoxification and the intermediary metabolism burden to skeletal muscle—where BCAAs serve as critical substrates. Muscle biopsy analyses have revealed that BCAA supplementation acutely reverses impaired mTOR signaling and excessive autophagic proteolysis, consistent with findings in alcoholic cirrhosis models expressing dysfunctional myostatin regulation [71]. Clinically, a 24-week BCAA supplementation in 106 cirrhosis patients resulted in notable improvements in several sarcopenia-related parameters, including handgrip strength, walking speed, abdominal muscle mass, triceps skinfold thickness, and overall muscle mass, with a corresponding reduction in the incidence of hepatic encephalopathy [81]. In a well-designed 12-week study of BCAA supplementation, seventeen patients (63%) demonstrated an overall improvement in muscle mass, with a notably higher response rate in the BCAA group compared to placebo (83.3% vs. 46.7%; *p* = 0.056). Additionally, the BCAA group showed a significant improvement in the Liver Frailty Index [82]. And several randomized controlled trials (RCTs) and systematic reviews have demonstrated that long-term oral BCAA supplementation improves event-free survival, reduces the incidence of hepatic encephalopathy, and enhances quality of life in cirrhotic patients with sarcopenia, though the effect on overall mortality remains inconsistent [59,76,83,84,85] (Table 2).

While many studies support the benefits of BCAA supplementation in cirrhosis–related sarcopenia, frailty, and hepatic encephalopathy, some trials and observational studies report limited or no benefit on key outcomes [86,87]. This underscores the need for further rigorously designed RCTs to clarify optimal indications and effective dosing strategies.

**Table 2 nutrients-17-03346-t002:** Systemic review and meta-analysis of BCAA in cirrhotic patients.

Author	Year	Number	Outcome	Etc.
Georgios Konstantis et al. [84]	2022	20 RCT	muscle mass, albumin level were improved prevention of clinical decompensation events	No effects for incidence of mortality
Anne M. van Dijk et al. [88]	2022	34 RCT, 5 Prospective, 13 Retrospective studies	event-free survival (*p* = 0.008; RR 0.61) improve overall survival (*p* = 0.05; RR 0.58)	Appears safe and might improve survival
Jia-Yu du et al. [83]	2022	9 RCT	Reduced the rate of complications (RR 0.70, *p* = 0.002) Albumin level improvement (SMD 0.26, *p* = 0.0002)	ALT, AST were ameliorated, and glucose level is increased.
Shoji Yokobori et al. [85]	2025	4 studies	disturbance of consciousness and mortality were not significantly different	outcomes were not significantly different between IV-BCAA treatment and placebo for acute HE.
Gluud LL et al. [59]	2017	16 RCT	Beneficial impact on HE (RR 0.73, 0.61–0.88)	No impact on mortality. (RR 0.88, 0.69–1.11)
Abdulrahman Ismaiel et al. [76]	2022	12	a significant improvement in skeletal muscle index (−0.347, *p* = 0.015) mid-arm muscle circumference (−1.273, *p* = 0.011).	No improvements were in handgrip, triceps subcutaneous fat.

ALT, Alanine aminotransferase; AST, Aspartate aminotransferase.

## 7. Conclusions

Malnutrition and sarcopenia are major prognostic factors in liver cirrhosis, driven by complex metabolic changes involving impaired protein synthesis, altered amino acid metabolism, and chronic inflammation. BCAA supplementation shows promise in improving muscle mass, ammonia detoxification, and clinical outcomes such as hepatic encephalopathy, but results across studies remain inconsistent. More well-designed trials are needed to define optimal indications and dosing. Additionally, underexplored mechanisms such as lipid metabolism, micronutrient interactions, and the gut–muscle–liver axis warrant further investigation.

Future therapeutic approaches should target not only energy and micronutrient imbalance but also the correction of impaired protein metabolism, which lies at the core of sarcopenia and poor prognosis in cirrhosis. Integrating nutritional, hormonal, and metabolic interventions may help restore protein homeostasis and improve survival in this population.

## Figures and Tables

**Table 1 nutrients-17-03346-t001:** Nutritional requirements according to guidelines.

	ESPEN [9]	AASLD [6]	EASL [21]
Total energy intake	35–40 kcal/kg/day	normal BMI: ≥35 kcal/kg/day obesity (BMI:30–40): 25–35 kcal/kg/day Morbid obesity (BMI > 40): 20–25 kcal/kg/day	35 kcal/kg/day
Total protein intake	1.2–1.5 g/kg/day	1.2–1.5 g/kg/day Sarcopenia: ≥1.5 g/kg/day	1.2–1.5 g/kg/day No recommendation for protein restriction

BMI, Body mass index.

## Data Availability

No new data were created or analyzed in this study.

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
