# Peer review of "Protein and Macronutrient Metabolism in Liver Cirrhosis: About Sarcopenia"

_nutrients, 2025, doi:10.3390/nu17213346_

Round 1

Reviewer 1 Report

Comments and Suggestions for Authors

Authors did a good job discriping current status of metabolic changes in patients with liver disease. The authors should consider including the Mendenhall manuscript as an example of the role of calories and protein intake and 6 month survival (Mendenhall CL, Hepatology, 1993).

Minor changes; Page 5 of 11 I would add the word potential to the word before the work improvements (Line 212)

Author Response

Reviewer # 1

Authors did a good job discriping current status of metabolic changes in patients with liver disease. The authors should consider including the Mendenhall manuscript as an example of the role of calories and protein intake and 6 month survival (Mendenhall CL, Hepatology, 1993).

Minor changes; Page 5 of 11 I would add the word potential to the word before the work improvements (Line 212)

Answer : Thank you for your thoughtful comment.
I have accepted your suggestion and cited it early in the manuscript. The word "potential" has also been added before "improvements" and is highlighted in yellow.

Reviewer 2 Report

Comments and Suggestions for Authors

The issue of nutrition in patients with chronic liver disease is an actual problem incompletely addressed by guidelines and a rapidly evolving field in the present.

The article is well structured and presents the mechanisms of becoming malnourished in patients with chronic liver disease. BCAA supplementation are recommended by guidelines as therapeutic adjunct for sarcopenia in cirrhosis. I suggest to detail the way BCAA work in patients with cirrhosis.

Lines 114-115 The sentence should rephrased. 

The Korean characters from table 2 should be removed.

Lines 197 - 204 - sarcopenia should be defined in the beginning of the article as the article is about managing this complication. 

Author Response

Reviewer #2

The issue of nutrition in patients with chronic liver disease is an actual problem incompletely addressed by guidelines and a rapidly evolving field in the present.

The article is well structured and presents the mechanisms of becoming malnourished in patients with chronic liver disease. BCAA supplementation are recommended by guidelines as therapeutic adjunct for sarcopenia in cirrhosis. I suggest to detail the way BCAA work in patients with cirrhosis.

Lines 114-115 The sentence should rephrased. 

And resulting insulin resistance like chronic inflammation.

 Answer : Thank you for pointing out this oversight. I have deleted the corresponding section according to your suggestion.”

The Korean characters from table 2 should be removed.

Answer : Thank you for pointing out this oversight. I have changed the section.

Lines 197 - 204 - sarcopenia should be defined in the beginning of the article as the article is about managing this complication. 

Answer : Your point is well taken. I have accepted your suggestion and moved the definition of sarcopenia to the early part of the Introduction, while keeping some of the related complications and risk factors at the end of the Sarcopenia section.
Originally, I intended to emphasize sarcopenia because many clinically important complications are closely related to protein-energy malnutrition.
However, in order to also address carbohydrate and lipid metabolism, the emphasis on sarcopenia was placed in the latter part of the manuscript.
In addition, based on comments from other reviewers, I have slightly revised the title and supplemented the content to further address the disturbances in protein metabolism. Thank you for kind recommendations.

Reviewer 3 Report

Comments and Suggestions for Authors

The manuscript presented by Han et al., on nutrition and chronic liver disease: macronutrient metabolism and sarcopenia, is very interesting as a general idea. However, the manuscript requires very important modifications. In this regard, I have the following comments.

I. Main comments:
1. The liver plays a significant role in macronutrient metabolism (carbohydrates, fatty acids, and amino acids). This hepatic function is very complex and has been extensively studied. Furthermore, in the face of various liver pathologies (hepatic steatosis, hepatitis, cirrhosis, hepatocellular carcinoma, etc.), research has been conducted and published. This is not clear in the manuscript.

2. When referring to sarcopenia, it would be logical to focus the review on protein and amino acid metabolism.

3. How do nutrient alterations (metabolic integration) relate to liver pathologies?

4. Considering the scope of the topic, I suggest that authors focus on a single nutrient, for example, amino acids and proteins, and briefly address the other nutrients.

5. Regarding amino acids, how is amino acid and protein synthesis altered? For example, leucine and isoleucine.

6. Does body weight influence hepatic metabolism of macronutrients?

7. Considering these changes, the manuscript could be revised.

Author Response

Reviewer #3

The manuscript presented by Han et al., on nutrition and chronic liver disease: macronutrient metabolism and sarcopenia, is very interesting as a general idea. However, the manuscript requires very important modifications. In this regard, I have the following comments.

  1. Main comments:
    1. The liver plays a significant role in macronutrient metabolism (carbohydrates, fatty acids, and amino acids). This hepatic function is very complex and has been extensively studied. Furthermore, in the face of various liver pathologies (hepatic steatosis, hepatitis, cirrhosis, hepatocellular carcinoma, etc.), research has been conducted and published. This is not clear in the manuscript.

Answer : Thank you for your insightful comments. We sincerely appreciate your valuable suggestions and completely agree with your points. Since metabolic alterations differ according to the stage of liver disease and systemic condition, I focused mainly on cirrhosis. In line with your helpful suggestion, I have added relevant mechanisms and revised the section on protein metabolism and sarcopenia accordingly.

  1. When referring to sarcopenia, it would be logical to focus the review on protein and amino acid metabolism.

Answer : Thank you for your thoughtful suggestion, with which I fully agree.

Accordingly, I have revised the title and expanded several sections in response to your and other reviewers’ comments.

  1. How do nutrient alterations (metabolic integration) relate to liver pathologies?

Answer : Thank you for your valuable suggestion. As you noted, metabolic alterations vary by disease stage. In line with recent evidence (Cell Reports Medicine, 2024), hepatic glucose production is influenced by intrahepatic inflammation and fibrosis rather than diabetes itself. Hence, this review focuses on cirrhosis and advanced liver disease, where these metabolic abnormalities are most pronounced.

  1. Considering the scope of the topic, I suggest that authors focus on a single nutrient, for example, amino acids and proteins, and briefly address the other nutrients.

Answer : We appreciate your careful review and thoughtful comments. The title has been revised accordingly, and additional content has been incorporated. And I added the contents about amino acid, protein synthesis results. Thank you.

  1. Regarding amino acids, how is amino acid and protein synthesis altered? For example, leucine and isoleucine.

Answer : We sincerely thank you for your valuable suggestions. Additional mechanisms related to leucine and isoleucine metabolism have been included and are highlighted in yellow in the revised version.

  1. Does body weight influence hepatic metabolism of macronutrients?

Answer : Thank you for your valuable comment. I agree that the concept of “body weight” can be somewhat ambiguous in patients with advanced cirrhosis. In such cases, ascites often contributes to an apparent increase in body weight, while simultaneously causing malnutrition through mechanisms such as reduced appetite, abdominal distension, dyspepsia, and impaired nutrient absorption.

If your comment refers to obesity or the overweight population within the context of steatotic liver disease, I would like to note that previous studies have suggested that even metabolically “healthy” obese individuals—those without overt cardiovascular or metabolic risk—may still represent a potential subgroup that progresses to steatotic liver disease. (Clin Mol Hepatol. 2023 Feb;29(suppl):S5-S16. doi: 10.3350/cmh.2022.0424.)

Therefore, rather than emphasizing the direct metabolic implications of obesity or excess body weight, I chose to focus this review on the stage of liver disease itself, particularly cirrhosis, where metabolic alterations and nutritional derangements are most clinically significant.

  1. Considering these changes, the manuscript could be revised.

Answer : Thank you for your detailed and thoughtful feedback. We have taken your comments into account and have made the necessary adjustments.

Round 2

Reviewer 3 Report

Comments and Suggestions for Authors

The authors responded to all my comments and questions. The manuscript was improved by the authors. I have no further questions. Therefore, the manuscript can be accepted.

Author Response

Thank you for pointing this out. We fully acknowledge your concerns and have revised the manuscript accordingly.